# Lipopeptides as rhizosphere public goods for microbial cooperation

Augustin Rigolet,[1] Anthony Argüelles Arias,[1] Adrien Anckaert,[1] Loïc Quinton,[2] Sébastien Rigali,[3,4] Deborah Tellatin,[3] Pierre Burguet,[2] Marc Ongena[1]

**ABSTRACT** Cyclic lipopeptides (CLPs) are key bioactive secondary metabolites produced by some plant beneficial rhizobacteria such as *Pseudomonas* and *Bacillus*. They exhibit antimicrobial properties, promote induced systemic resistance in plants, and support key developmental traits, including motility, biofilm formation, and root colonization. However, our knowledge about the fate of lipopeptides once released in the environment and especially upon contact with neighboring rhizobacteria remains limited. Here, we investigated the enzymatic degradation of *Bacillus* and *Pseudomonas* cyclic lipopeptides by *Streptomyces*. We observed that *Streptomyces venezuelae* is able to degrade the three lipopeptides surfactin, iturin, and fengycin upon interaction with *Bacillus velezensis in vitro* and *in planta* according to specific mechanisms. *S. venezuelae* was also able to degrade the structurally diverse sessilin-, tolaasin-, orfamide-, xantholisin-, and putisolvin-type lipopeptides produced by *Pseudomonas,* indicating that this trait is likely engaged in the interaction with various competitors. Furthermore, the degradation of CLPs is associated with the release of free amino and fatty acids used by *Streptomyces* to sustain growth. Additionally, we hypothesize that lipopeptide-producing rhizobacteria and their biocontrol potential are impacted by the degradation of their lipopeptides as observed with the polarized motility of *B. velezensis*, avoiding the confrontation zone with *Streptomyces* and the loss of antifungal properties of degraded iturin.

**IMPORTANCE** Here, we provide new insights into the possible fate of cyclic lipopeptides as prominent specialized metabolites from beneficial bacilli and pseudomonads once released in the soil. Our data illustrate how the *B. velezensis* lipopeptidome may be enzymatically remodeled by *Streptomyces* as important members of the soil bacterial community. The enzymatic arsenal of *S. venezuelae* enables an unsuspected extensive degradation of these compounds, allowing the bacterium to feed on these exogenous products via a mechanism going beyond linearization, which was previously reported as a detoxification strategy. As soils are carbon-rich and nitrogen-poor environments, we propose a new role for cyclic lipopeptides in interspecies interactions, which is to fuel the nitrogen metabolism of a part of the rhizosphere microbial community. *Streptomyces* and other actinomycetes, producing numerous peptidases and displaying several traits of beneficial bacteria, should be at the front line to directly benefit from these metabolites as "public goods" for microbial cooperation.

**KEYWORDS** *Bacillus velezensis*, *Streptomyces venezuelae*, *Pseudomonas*, BSMs, cyclic lipopeptides, enzymatic degradation, competition, interaction, foraging, feeding, rhizosphere

Cyclic lipopeptides (CLPs) represent a prominent and structurally heterogeneous class of molecules among the broad spectrum of small bioactive secondary metabolites

Address correspondence to Augustin Rigolet, arigolet@uliege.be, or Marc Ongena, marc.ongena@uliege.be.

The authors declare no conflict of interest.

(BSMs) produced by some plant beneficial rhizobacteria such as *Pseudomonas* and *Bacillus* (1, 2). These amphiphilic compounds consist of a partly or fully cyclized oligopeptide linked to a single fatty acid. They have been shown to inhibit the growth of a large range of phytopathogens and elicit immune responses in the host plant, leading to an induced systemic resistance against infection by microbial pathogens (3, 4). These traits are largely responsible for the biocontrol potential of some CLP-producing isolates used to reduce plant diseases in sustainable agriculture (3). From an ecological perspective, antimicrobial CLPs also contribute to the weaponry developed by these plant-associated bacteria to harm or kill microbial competitors in the densely populated rhizosphere niche. Moreover, CLPs support key developmental traits such as motility, biofilm formation, or root colonization (2, 3, 5).

CLPs are quite efficiently produced both *in vitro* and under natural conditions, and substantial amounts are presumably released in the surrounding environment (5–7). These metabolites are considered as chemically stable compounds due to the closed structure of the peptide moiety, the alternation of D- and L-amino acids, and the incorporation of non-proteinogenic residues (2). These molecules may thus accumulate in the rhizosphere, impact microbial interactions, and modulate the composition of soil microbiomes. However, some recent studies reported instability of CLPs in the soil or in synthetic communities (8–10), but the mechanisms underlying CLP degradation as well as the possible ecological outcomes resulting from the phenomenon are still poorly described.

In this work, we wanted to investigate the possible degradation of CLPs by *Streptomyces* as soil representative and using *Streptomyces venezuelae* as model species known for its metabolic robustness, behavioral plasticity, and extensive enzymatic arsenal (11). We first confronted the natural isolate *Streptomyces venezuelae* ATCC 10712 (Sv) to *Bacillus velezensis* GA1 (Bv) strain, an archetypical root-associated isolate that efficiently co-produces surfactin, iturin, and fengycin as the three lipopeptide families typical of the *Bacillus subtilis* group (12, 13). Bacteria were inoculated at a distance on gelified root exudate-mimicking medium (REM) designed to reflect the nutritional context of the rhizosphere (Fig. 1A). Sv colonies were phenotypically similar in interaction compared to monoculture, while Bv colonies displayed a polarized growth and altered motility close to Sv (Fig. 1A; Fig. S1). Ultraperformance liquid chromatography (UPLC)-quadrupole time-of-flight-mass spectrometry (MS) metabolite profiling of the compounds extracted from the agar in the confrontation zone revealed a decrease in the abundance of the three *Bacillus* CLP families compared with monocultures (Fig. 1B; Fig. S1B), along with the accumulation of their cognate linearized forms eluting earlier (lower apparent hydrophobicity, Fig. 1B). These linearized products were identified based on mass increment of 18 Da, resulting from the hydrolysis of the peptide ring, and further confirmed by tandem mass spectrometry (MS/MS) structure elucidation (Fig. 1C; Fig. S2 to S4). Interestingly, additional ion species corresponding to shorter CLP fragments of surfactin (loss of the fatty acid from the linear form), iturin (loss of asparagine in position 3), and fengycin (loss of the terminal isoleucine) were also detected in the confrontation zone but not in Bv monoculture (Fig. 1B bottom panels, MS/MS spectra in Fig. S2 to S4). We next confronted Sv and the GFP-tagged GA1 upon colonization of tomato roots in a setup better mimicking rhizosphere conditions. When inoculated alone, Bv readily colonizes roots as biofilm-structured colonies (Fig. 1D) and efficiently forms the three lipopeptides in their native cyclic structure as revealed by UPLC-MS analysis of rhizosphere extracts (Fig. 1E). Upon co-inoculation with Sv, which forms mycelial pellets along the roots, there is no spatial exclusion of Bv, which still colonizes roots and secretes lipopeptides in substantial amounts (Fig. 1D and E). However, as for plate confrontation, a high proportion of linear iturins and surfactins (but not fengycins), along with surfactin fragments, were observed in rhizosphere extracts, indicating that some degradation of Bv CLPs by Sv also occurs under the natural context of root co-colonization (Fig. 1E).

Based on these data, we further explored the Sv-mediated alteration of *Bacillus* CLPs and investigated the degradation process beyond linearization by using purified CLPs

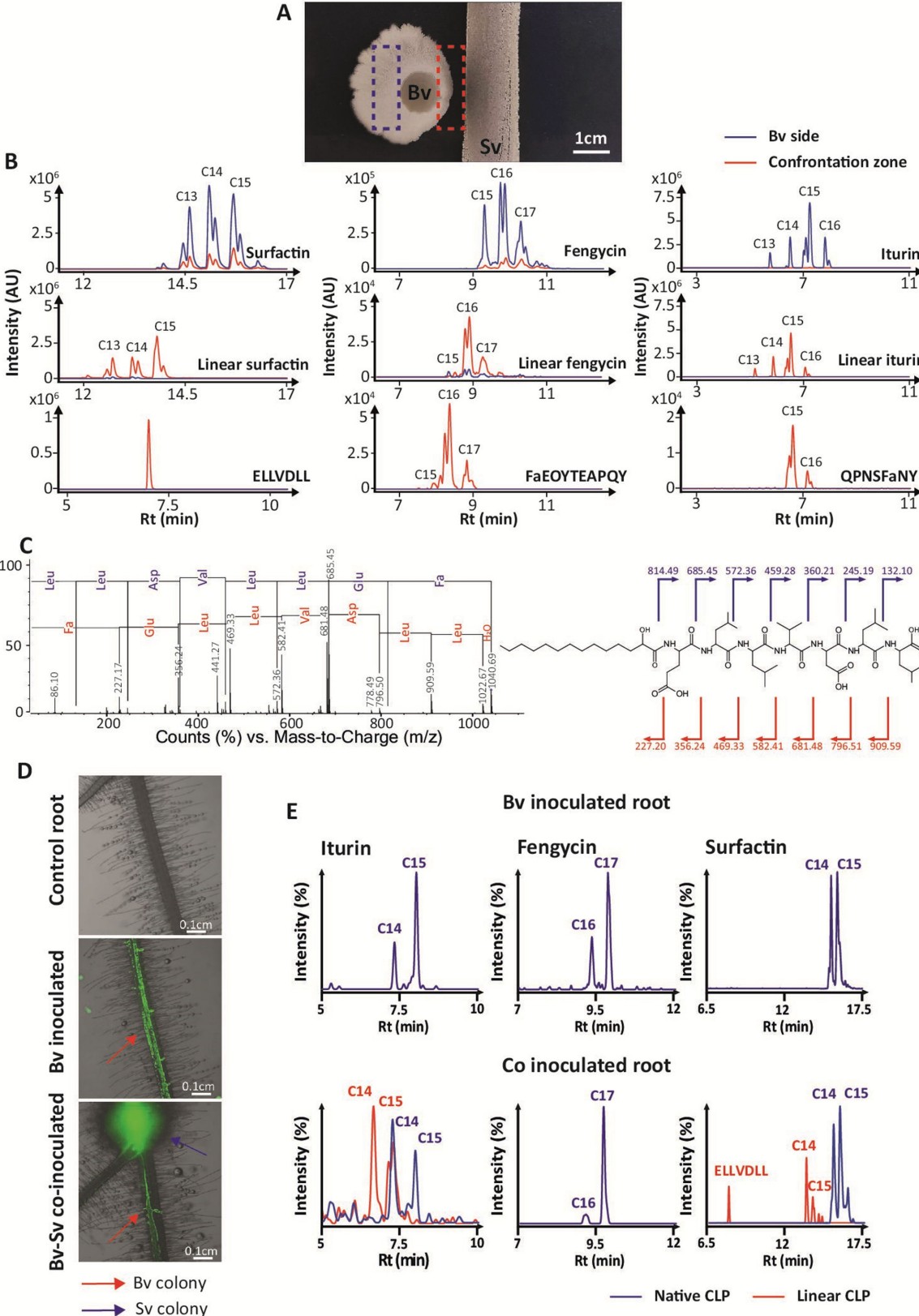

**FIG 1** *Streptomyces venezuelae* ATCC 10712 linearizes *Bacillus* lipopeptide surfactin, iturin, and fengycin upon interaction with *Bacillus velezensis* GA1. (A) Picture of the interaction between *B. velezensis* GA1 (Bv, left side) and *S. venezuelae* ATCC 10712 (Sv, right side) on plate. Dashed rectangle represents the sampling areas used for metabolite extraction. The picture is representative of three biological replicates. (B) UPLC-ESI-MS EIC of canonical (top panels), linear (middle panels),

**FIG 1** (Continued)

and degradation products (bottom panels) of surfactin, iturin, and fengycin extracted from agar in the interaction zone in between Sv and Bv (confrontation zone, in red) and on the Bv side (in blue). The EICs are merged chromatograms of the $[m + H]^+$ monoisotopic adducts of the main variants of each CLP. "Cn" represents the number of carbon of the fatty acid of the main CLP variant detected in each peak. Chromatograms are representatives of three biological replicates. Mean peak areas of the replicates of the different CLPs and linearized CLPs are shown in Fig. S1. (C) LC-ESI-MS/MS spectra of linear surfactin C14 and corresponding structure. Blue and red clippers and arrows represent the y- and b-ions. (D) Merged bright field and green fluorescence stereomicroscopic photos of tomato roots, non-inoculated (control root, top picture), inoculated with GA1 GFPmut3-tagged (Bv inoculated, middle picture), and co-inoculated with GA1 GFPmut3 and *S. venezuelae* ATCC 10712 (Bv-Sv co-inoculated, bottom picture). Pictures are representatives of four biological replicates. *S. venezuelae* is naturally fluorescent, but its colonies are distinguished from *Bacillus* colonies by their phenotypes (*Streptomyces* forms mycelial pellets and *Bacillus* forms biofilm structures). (E) UPLC-ESI-MS EIC of canonical (blue) and linear (red) surfactin, fengycin, and iturin extracted from the surrounding of the tomato roots inoculated with Bv (top panels) and co-inoculated with Bv and Sv (bottom panel). Chromatograms are representatives of four biological replicates. EIC, extracted ion chromatogram; ESI, electrospray ionization; FA, fatty acid; LC, liquid chromatography.

supplemented with Sv cell-free supernatant (CFS) in a time-course experiment combined with feature-based molecular networking (FBMN). For each CLP, FBMN identified multiple degradation products generated in the presence of Sv CFS, including those detected in confrontation assays and *in planta* (Fig. 2A through C, MS/MS spectra in Fig. S2 to S4). Based on the fragments identified by FBMN and time-course monitoring of their occurrence (Fig. 2D; Fig. S5), we propose a degradation mechanism specific for each CLP characterized by the sequential generation of linearized lipopeptides followed by truncated fragments (Fig. 2A through C). In a similar setup, we observed that several other strains of *Streptomyces* were able to degrade Bv CLPs, indicating that this trait is not unique to Sv (Fig. S6). Likewise, we also tested Sv CFS for its ability to break down *Pseudomonas* CLPs representative of some of the main classes produced by soil-borne species (1). Albeit to different degrees, sessilin, tolaasin, orfamide, xantholisin, and putisolvin were all degraded (Fig. S7 to S11), indicating that Sv may target a broad range of structurally diverse CLPs that the bacterium is likely to encounter in the soil. In most cases, degradation initiates with the opening of the peptide cycle followed by iterative degradation of the linear form, associated with the release of free fatty or amino acids. These mechanisms suggest the involvement of several enzymes secreted by Sv, including esterase or endo-proteases for linearization and exo-proteases to further degrade the peptide. The enzymatic nature of the degradation was confirmed as heat-treated CFS of Sv completely loses its degradation activity (Fig. S12), and comparative proteomics revealed the presence of several secreted proteases and amino acid/oligopeptide transporters specific to active CFS of Sv (Table S3).

Therefore, we hypothesized that Sv may catabolize those exogenous CLPs and use them as nutritional sources. Indeed, we observed Sv can sustain some growth on gelified CLP-free Bv supernatant (GA1 Δ*sfp*, mutant unable to produce the *sfp*-dependent metabolites: CLPs, PKs and the siderophore bacillibactin) but grows significantly more when cultivated on gelified CLP-containing CFS of Bv mutants [GA1 Δ*baeJ-mlnA-dfnA*, mutant unable to produce Bv antibacterial polyketides bacillaene, difficidin, and macrolactin, known for their toxicity toward *Streptomyces* (14)] (Fig. 2E and F). Extraction of the metabolites from the Sv cultures grown on CLP-containing conditions reveals the presence of degradation products of both iturin, surfactin, and fengycin, further indicating that increased growth is driven by CLP catabolism (Fig. 2G).

We propose that the ability of Sv to degrade CLPs and feed on them adds a new facet to the implications of CLPs degradation by *Streptomyces*, where linearization of surfactin was previously reported as a detoxification mechanism deployed by *Streptomyces* to counteract the inhibitory effect of this CLP on aerial mycelium formation (15). As soils are carbon-rich and nitrogen-poor environments, exogenous CLP degradation may thus represent a foraging strategy for *Streptomyces* to access alternative sources of nutriments directly emanating from diverse microbial competitors. We assume that, in this peculiar nutritional context, such a foraging strategy may be part of a larger microbial cooperation between CLP producers and consumers. CLPs may actually also represent public goods that fuel the nitrogen metabolism of a part of the rhizosphere

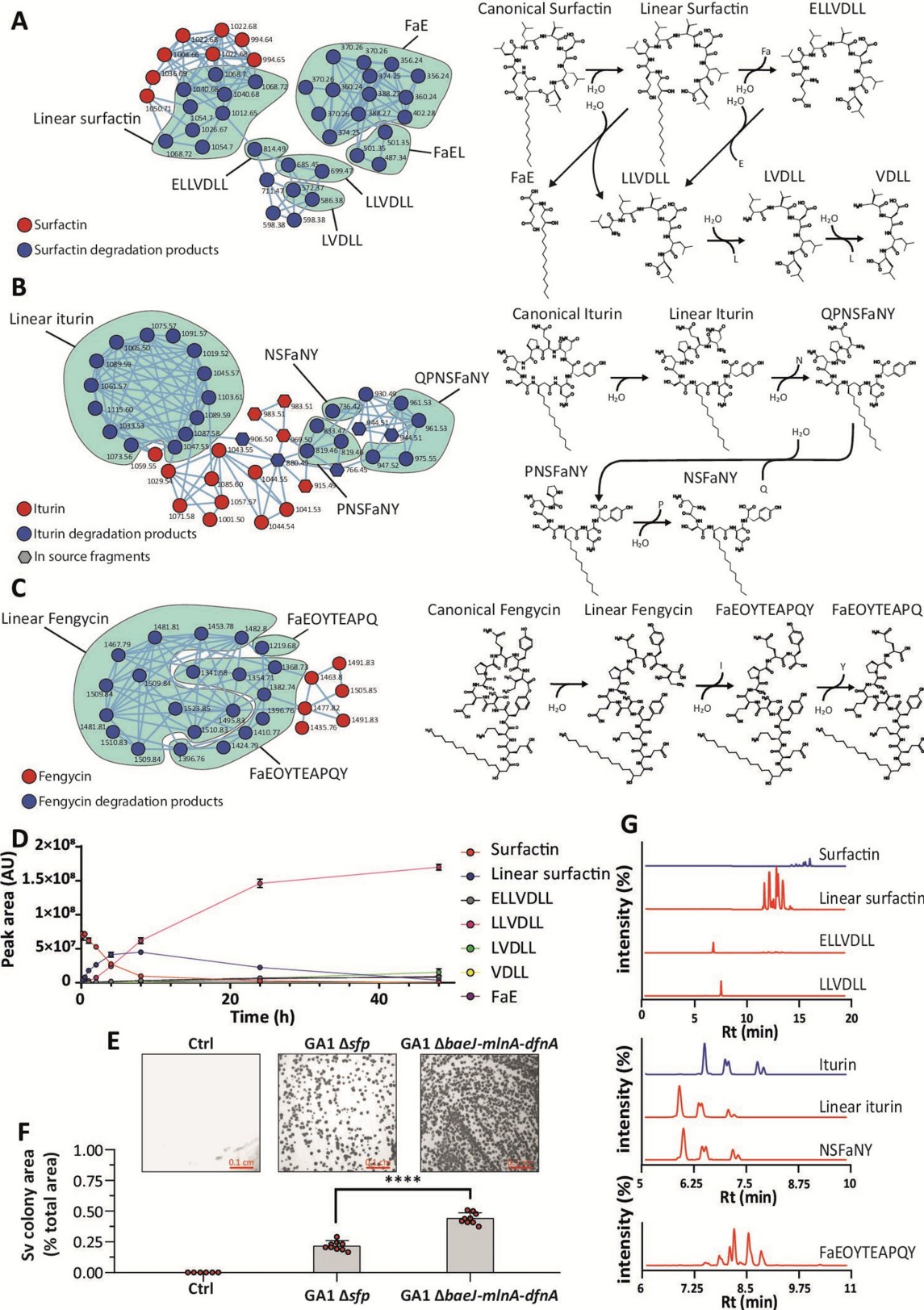

**FIG 2** *S. venezuelae* degrades *B. velezensis* CLPs and feeds on it. (A–C) Feature-based molecular networking of the degradation products of *Bacillus* CLP surfactin (A), iturin (B), and fengycin (C) generated by Sv and proposed degradation mechanisms of each CLP. Pure CLPs (100 µM) were incubated for 24 h at 30°C with 4% (vol/vol) of Sv CFS. MS/MS spectra are available in Fig. S2 to S4. The degradation mechanisms are proposed based on the fragments detected. Summary of

**FIG 2** (Continued)

the identified features is available in Table S4. (D) Time-course degradation of surfactin upon incubation with Sv CFS 4% (vol/vol). (E) Pictures of the Sv colonies grown on gelified REM salt solution (Ctrl), GA1 *ΔbaeJ-mlnA-dfnA*, and GA1 *δsfp* supernatant. The *sfp* gene encodes for a 4'-phosphopantetheinyl transferase required for the activation of the synthesis of non-ribosomal peptides and PKs. The mutant GA1 *Δsfp* is unable to synthesize the Sfp-dependent BSMs: the CLPs (surfactin, iturin, and fengycin), the PKs (bacillaene, difficidin, and macrolactin), and the siderophore bacillibactin. The mutant GA1 *ΔbaeJ-mlnA-dfnA* is unable to synthesize the PKs bacillaene, macrolactin, and difficidin. Both strains were grown on iron sufficient medium to repress bacillibactin production. We used the mutants repressed in the synthesis of the PKs (*ΔbaeJ-mlnA-dfnA* and *Δsfp*) as Bv PKs inhibit Sv growth at high concentrations (i.e., when Sv grows on Bv CFS). Pictures are representatives of six replicates. (F) Relative summed colony area of Sv upon growth on gelified REM salt solution (Ctrl) ,GA1 *ΔbaeJ-mlnA-dfnA*, and GA1 *Δsfp* supernatants. Picture areas used for colony area measurement = 0.025 cm$^2$. Each dot represents a biological replicate ($n$ = 9). Statistical significance was calculated using Mann-Whitney test. ****$P$ < 0.0001. (G) LC-ESI-MS EIC chromatograms of *Bacillus* CLPs iturin, surfactin, and fengycin and the corresponding degradation products in Sv cultures grown on gelified GA1 *ΔbaeJ-mlnA-dfnA* CFS. The chromatograms are representatives of two biological replicates. PK, polyketide.

microbial community. This subset of the community includes *Streptomyces* and other actinomycetes which have not only a huge arsenal of peptidases but also several plant-beneficial traits contributing to the host plant health together with, and sustained by, CLP-producing bacilli and pseudomonads.

Additionally, CLPs are key multifunctional BSMs whose biocontrol-associated activities often involve membrane perturbation and pore formation (16). This CLP-membrane interaction is enabled by the peculiar amphiphilic three-dimensional structures of those CLPs (17, 18). However, since the degradation alters their structures, it is likely associated with a loss of function. This was confirmed via additional experiments where we observed that digested iturin loses its antifungal activities against phytopathogenic fungi *Fusarium* and *Botrytis in vitro* (Fig. S13). Likewise, linearization of surfactin has been associated with an almost full loss of its potential to stimulate immune-related defense responses in tobacco cells (17, 19). Nonetheless, the impact degradation has on the biocontrol activities of other CLPs and, more globally, on the biocontrol potential of CLP-producing rhizobacteria deserves further investigation.

Furthermore, CLP degradation also possibly hampers the ecological fitness of the producers as it may impact some phenotypical traits favored by CLPs such as biofilm formation, motility, and root colonization. The polarized motility of Bv in the presence of Sv observed in Fig. 1A may indeed result from the degradation of surfactin in the confrontation zone since the potential of this CLP to stimulate motility is structure dependent (20). Yet, a more comprehensive evaluation of the impact of CLP degradation on the various Bv developmental traits they promote and that contribute to environmental fitness deserves further investigations.

Finally, The degradation of CLPs increases the chemical space resulting from the interaction as it generates numerous degradation products. It has been reported that the degradation by *Paenibacillus* of the lipopeptide syringafactin produced by *Pseudomonas* generates products toxic to their common amoeba predators (21) and that the degradation of surfactin, also by *Paenibacillus*, serves as a deterrent or a territory marker in the interaction with *B. subtilis* (22). Likewise, unsuspected bioactivities may rise from the degradation of *Bacillus* CLPs by *Streptomyces*.

## MATERIALS AND METHODS

### Strains and cultures conditions

Strains used are listed in Table S1.

All experiments with *S. venezuelae* ATCC 10712 and other *Streptomyces* strains were inoculated with spore suspensions (10$^7$ spores/mL). *Streptomyces* spores were recovered from soya flour mannitol (SFM) medium plate (soy flour 20 g/L, mannitol 20 g/L, agar 20 g/L, and tap water 1 L; pH 7.2) and stored at −80°C in peptone water (peptone 10 g/L, NaCl 5 g/L) supplemented with glycerol 25% (vol/vol). Spore concentrations were measured with Bruker cells.

All *B. velezensis* GA1 wild type and mutants and phytopathogenic bacteria were routinely precultured overnight in REM {0.5 L of all media [0.685 g of $KH_3PO_4$, 21 g of 3-(N-morpholino)propanesulfonic acid (MOPS), 0.5 g of $MgSO_4 \cdot 7H_2O$, 0.5 g of KCl, and 1.0 g of yeast extract]; 100 µL of the trace solution [0.12 g of $Fe_2(SO_4)_3$, 0.04 g of $MnSO_4$, 0.16 g of $CuSO_4$, and 0.4 g of $Na_2MoO_4$/10 mL]; and 0.5 L of tobacco medium [2.0 g of glucose, 3.4 g of fructose, 0.4 g of maltose, 0.6 g of ribose, 4.0 g of citrate, 4.0 g of oxalate, 3.0 g of succinate, 1.0 g of malate, 10 g of fumarate, 1.0 g of casamino acids, 2.0 g of $(NH_4)_2SO_4$/liter, pH 7.0]} as described by reference (23) at 30°C. After being washed three times in REM (cells were collected, centrifuged at 10,000 rpm for 1 min and resuspended in fresh REM), bacterial suspensions were set at proper $OD_{600\ nm}$ specified for each experiment and used for the experimental setup.

*Pseudomonas* strains were routinely precultured in casamino acid (CAA) liquid medium (10-g/L casamino acid, 0.3-g /L $K_2HPO_4$, 0.5-g/L $MgSO_4$, and pH 7.0) at 30°C. After being washed three times in casamino acid liquid medium (cells were collected, centrifuged at 10,000 rpm for 1 min, and resuspended in fresh medium), bacterial suspensions were set at proper $OD_{600\ nm}$ and used for the experimental setup.

## Construction of *Bacillus* knock-out mutant strains

The triple mutant GA1 *ΔbaeJ-dfnA-mlnA* was constructed from the mutant strain GA1 *ΔbaeJ-dfnA* from (13) . On this mutant, the *mlnA* gene was deleted by allelic replacement using a mutagenesis cassette containing a phleomycin resistance gene (50 µg/mL) flanked by 1 kb of the upstream region and 1 kb of the downstream region of the targeted gene. Mutagenesis cassettes were constructed by overlap PCR as described by reference (24). The primers used were

UpF: CGGAAAAACCGTTTCAAAAA
UpR: CAGGAAACAGCTATGACTTTTAAAATTGTCATTTACTCTAAGCA
DwF: GTAAAACGACGGCCAGTCTAAGGCGCAGATTGGATA
DwR:TGTACCTGTGCCATGTGCTT

The recombination cassette was introduced in *B. velezensis* GA1 *ΔbaeJ-dfnA* by inducing natural competence using a method adapted from reference (25). Briefly, after an initial preculture in Luria-Bertani (LB) medium at 37°C (160 rpm) during at least 6 h, cells were washed twice with peptone water. One microgram of the recombinant cassette was added to the GA1 cell suspension adjusted to an $OD_{600\ nm}$ of 0.01 into MMG liquid medium [19-g/L $K_2HPO_4$ anhydrous, 6-g/L $KH_2PO_4$, 1-g/L $Na_3$ citrate anhydrous, 0.2-g/L $MgSO_4$ $7H_2O$, 2-g/L $Na_2SO_4$, 50-µM $FeCl_3$ (filtrated on 0.22-µm pore size filters), 2-µM $MnSO_4$, 8-g/L glucose, 2-g/L L-glutamic acid, pH 7.0]. After 24 h of incubation at 37°C with shaking, double-crossover events were selected on LB plates supplemented with the appropriate antibiotic. The gene deletion was confirmed by PCR analysis using primers CGGAAAAACCGTTTCAAAAA and GTCCGGCGTAGAGGATCTG and by liquid chromatography (LC)-mass spectrometry (MS) (loss of macrolactin production).

## Confrontation experiments

Confrontation assays were performed on square Petri dishes (12 × 12 cm) with 40 mL of REM solid medium at 26°C (REM supplemented with 14 g/L of agar).

*S. venezuelae* ATCC 10712 was inoculated as stripes (1 × 12 cm) in the middle of the plate with 40 µL of spore suspension ($10^7$ spores/mL) and spread with a cotton swab. Next, *B. velezensis* GA1 cells were collected from fresh precultures as described and adjusted to $OD_{600\ nm}$ 0.1. Then, 5 µL of *B. velezensis* GA1 suspension was spotted at 1 cm of the *Streptomyces* line. Control plates were done following the same procedure without the inoculation of either *Bacillus* or *Streptomyces*. Plates were then incubated for 3 days at 26°C in the dark. Pictures of the plates were then captured using a CoolPix camera (NIKKOR ×60 wide optical zoom extra-low dispersion vibration reduction 4.3–258.0 mm 1:33 to 6.5).

## *In planta* experiments

For the *in planta* studies, tomato seeds (*Solanum lycopersicum* var. Moneymaker) were sterilized following the protocol described by (26). Briefly, tomato seeds were primarily sterilized in 70% ethanol (vol/vol) by gently shaking for 2 min. Further, the ethanol was removed and the seeds were added to 50 mL of the sterilization solution [4.5 mL of bleach containing 9.5% (vol/vol) of active chlorine, 0.01 g of Tween 80, and 45.5 mL of sterile water] and gently shaken for 10 min. Seeds were thereafter washed 10 times with water to eliminate sterilization solution residues. Sterilized seeds were then placed on square Petri dishes (12 × 12 cm) (five seeds per plate) containing Hoagland solid medium {14-g/L agar, 5 mL of stock 1 [EDTA 5.20 mg/L, $FeSO_4·7H_2O$ 3.90 mg/L, $H_3BO_3$ 1.40 mg/L, $MgSO_4·7H_2O$ 513 mg/L, $MnCl_2·4H_2O$ 0.90 mg/L, $ZnSO_4·7H_2O$ 0.10 mg/L, $CuSO_4·5H_2O$ 0.05 mg/L, 1 mL in 50-mL stock 1, $NaMoO_4·2H_2O$ 0.02 mg/L 1 mL in 50-mL stock 1], 5 mL of stock 2 [$KH_2PO_4$ 170 mg/L], 5 mL of stock 3 [$KNO_3$ 316 mg/L, $Ca(NO_3)_2·4H_2O$ mg/L], pH 6.5} and were placed in the dark to germinate for 3 days. Afterward, germinated seeds were inoculated with 2 µL of the culture ($OD_{600\ nm}$ 0.1) of *B. velezensis* GA1-GFP and 2 µL of spore suspension ($10^7$ spores/mL) of *S. venezuelae* ATCC 10712 or with both GA1 and ATCC 10712 (co-inoculation), and grown at 22°C under a 16-/8-h day/night cycle with constant light for 7 days.

For BSM production analysis in *in planta* conditions, an agar part (1 × 1 cm) near the tomato roots was cut and weighted. Extraction and UPLC-MS analysis of the metabolites were then performed with the same protocol as described for the confrontation assays.

Stereomicroscopic pictures of inoculated tomato roots were taken with a Nikon SMZ1270 stereomicroscope (Nikon, Japan) equipped with a Nikon DS-Qi2 monochrome microscope camera and a DS-F 2.5 × F-mount adapter ×2.5. Pictures were captured in the bright field channel and green widefield fluorescence (emission 535 nm, excitation 470 nm) with an ED Plan 2×/WF objective at an exposure time of 40 ms. NIS-Element AR software (Nikon) was used to generate merged bright field and green fluorescence. Background and root green autofluorescence were removed by adjusting the LUTs (3388–6638).

## Metabolite extraction and sample preparation

The extraction of the metabolites from the agar was performed as follows. For the confrontation experiment, we precisely sampled the area of agar (2 × 1 cm) near the colony of *Bacillus* or *Streptomyces* (the confrontation zone is 1 cm wide) with standardized plugs. For the *in planta*, we sampled the agar area (1 × 1 cm) near the tomato roots. In both cases, the agar samples were transferred to Eppendorf tubes and placed for 24 h at −20°C. Then, the agar was thawed at room temperature and centrifuged for 10 min at 13,000 rpm. The supernatants were then collected and filtered (0.22-µm pore size filters) before UPLC-MS analysis.

## Generation of cell-free supernatants

The generation of CFSs of *B. velezensis* wild-type and mutant strains (Δ*sfp* or GA1 Δ*baeJ-dfnA-mlnA*) was performed as follows. We inoculated 250 mL flasks containing 50 mL REM culture with at initial $OD_{600\ nm}$ 0.02 with cells from fresh precultures as previously described. We then incubated the cultures for 24 h at 30°C with continuous orbitally shaking (180 rpm). Next, the cultures were centrifuged at 5,000 rpm at room temperature for 20 min. The supernatants were further filter-sterilized (0.22-µm pore size filters) and stored at −20°C until further use.

Generation of *Pseudomonas* CFS was performed as follows: we inoculated 250-mL flasks containing 50-mL CAA culture with at initial $OD_{600\ nm}$ of 0.02 with cells from fresh precultures as described. We then incubated the culture for 48 h at 26°C with continuous orbitally shaking (180 rpm). Next, the cultures were centrifuged at 5,000 rpm at room temperature for 20 min. The supernatants were further filter-sterilized (0.22-µm pore size filters) and stored at −20°C until further use.

*S. venezuelae* cultures were performed on ISP2 medium at 28°C (yeast extract 4 g/L, malt extract 10 g/L, glucose 4 g/L, agar 20 g/L; pH 7.3). On 12 × 12 cm Petri square plates, three stripes (1 × 12 cm) of spores were inoculated with a cotton swab [40 µL of spores suspension ($10^7$ spores/mL) each], spaced by 3 cm each. The plates were left for 7 days of incubation. Next, the agar media was recovered and placed in Falcon tubes at −20°C for 24 h. Then, the agar media were defrosted at room temperature and centrifuged (8,000 rpm for 20 min). Finally, the supernatant leaked from the agar was collected, filtered sterilized (0.22-µm pore size filters) and stored at −20°C until further use. The heat-treated Sv supernatant was generated by incubating an aliquot of the cell-free supernatant of Sv for 10 min at 98°C. The supernatant was then filtered (0.22-µm pore size filters) and stored at −20°C until further use.

### *In vitro* degradation of CLP assays by *S. venezuelae*

The degradation assays of *Bacillus* CLPs by *S. venezuelae* were performed as follows: 500-µL solutions of pure surfactin, iturin, or fengycin (40 µM) was supplemented by 4% (vol/vol) of *S. venezuelae* supernatant (or heat-treated supernatant) prepared as previously described. Then the solutions were incubated for 24 h at 30°C with continuous shaking (180 rpm). Next, the solution were centrifuged (1 min at 10,000 rpm), filtered (0.22-µm pore size filters), and analyzed by UPLC-MS.

The degradation assays of *Pseudomonas* CLPs were performed using CFS generated as described above. They were supplemented by 4% (vol/vol) of *S. venezuelae* CFS prepared as described previously. Next, the solutions were incubated for 24 h at 30°C with continuous shaking (180 rpm). Then, the solution were centrifuged (1 min at 10,000 rpm), filtered (0.22-µm pore size filters), and analyzed by UPLC-MS.

For the CLP degradation kinetic experiments, surfactin, iturin, and fengycin solutions supplemented with *S. venezuelae* supernatant were prepared following the same protocol. Twenty microliters of solution was sampled at each time point and directly mixed with 80 µL of acetonitrile to stop enzymatic degradation. Finally, they were stored at −20°C until analysis by UPLC-MS.

### UPLC-MS analysis

All UPLC-MS analyses were performed using an Agilent 1290 Infinity II coupled with a diode array detector and a mass detector (Jet Stream ESI-Q-TOF 6530) in positive mode with the parameters set up as follows: capillary voltage of 3.5 kV, nebulizer pressure of 35 lb/in², drying gas of 8 L/min, drying gas temperature of 300°C, flow rate of sheath gas of 11 L/min, sheath gas temperature of 350°C, fragmentor voltage of 175 V, skimmer voltage of 65 V, and octopole radiofrequency of 750 V. Accurate mass spectra were recorded in the *m/z* range of 300–1,700. For untargeted MS/MS, we used the same MS1 parameters as described. We added MS2 untargeted acquisition mode with the parameters as follows: MS/MS range 50–1,700 *m/z*; MS/MS scan rate 3 spectra/s; isolation width MS/MS medium (approx. 4 AMU), Decision Engine Native; fixed collision energies of 25 and 40 V for surfactin, 50 V for iturin, and 60 V for fengycin experiments; precursor selection: three for the surfactin experiment, four for the iturin and fengycin experiments, threshold 1,500 (Abs), isotope model common, active exclusion after two spectra and release after 0.5 min, sorting the precursors by charge state then abundance (charge state preference 1). For targeted MS/MS, we used the same MS1 parameters MS/MS range of 50–1,700 *m/z* or 50–3,200 (when required for *Pseudomonas* CLPs with mass >1,700 Da), MS/MS scan rate 3 spectra/s, isolation width MS/MS narrow (approx. 1.3 amu), and fixed collision energies 20, 40, and 60V. In all experiments, a C18 Acquity UPLC ethylene bridged hybrid column (2.1 mm × 50 mm × 1.7 µm; Waters Corporation, Milford, MA, USA) was used at a flow rate of 0.6 mL/min and a temperature of 40°C. The injection volume was 20 µL, and the diode array detector scanned a wavelength spectrum between 190 and 600 nm. Unless otherwise mentioned, a gradient of acidified water (0.1% formic acid) (solvent A) and of acidified acetonitrile (0.1% formic acid) (solvent B) was used as mobile phase with a constant flow rate of 0.6 mL/min, starting

at 10% B and rising to 100% B in 20 min. Solvent B was kept at 100% for 4 min before going back to the initial ratio. MassHunter Workstation (version 10.0) and ChemStation software were used for data collection and analysis. For untargeted analysis of iturin and fengycin degradation products, we used the same solvent and flow rate, starting at 10% B to 20% in 2 min, then rising to 50% B at 14 min and 100% B at 25 min, followed by 100% B and 5 at 10% B at 6 min.

## MZmine-GNPS analysis

MZmine 3 parameters used in this study are listed in Table S3. Feature lists were then exported and submitted to Global Natural Products Social Molecular Networking (GNPS). GNPS analysis of each CLP was performed with the following parameters: quantification table source: MZmine, precursor ion mass tolerance: 0.02 Da, fragment ion mass tolerance: 0.02 Da, min pairs cos: 0.5 for iturin and fengycin and 0.6 for surfactin, minimum matched fragment ions: 4, maximum shift between precursors: 500, network TopK: 10, and maximum connected component size: 100. All the other parameters were set as defaults. GNPS networks were then exported to cytoscape. Nodes corresponding to canonical CLP were identified based on the exact mass, the retention time, the presence in control CLP samples (without Sv supernatant treatment) and confirmation with the MS/MS spectra. Conversely, degradation products were identified as connected to canonical CLP and accumulation in CLP samples treated with Sv. The structures of the fragments were then elucidated with the MS/MS spectra.

The GNPS job IDs are, for surfactin: ID = f74804937ce14ab4b211e4e2ee3647a0, for iturin: ID = a2ac85a8fe49439fbf8235d514d31b1a, and for fengycin: ID = 336c4c73ab6642f68787a173cf3ca719.

## Growth on CLPs assays

The ability of *Streptomyces* to grow on *B. velezensis* CLPs was determined on 48-well microplates. Each well was filled with 500 µL of agar solution (40 g/L agar) and 500 µL of cell-free supernatant of *B. velezensis* GA1, GA1 *Δsfp*, or GA1 *ΔbaeJ-dfnA-mlnA*. Wells were inoculated with 5 µL of spore suspensions of *Streptomyces* (OD$_{600\ nm}$ 0.1). The microplates were incubated for 3 days at 28°C.

Stereomicroscopic pictures of inoculated tomato roots were taken with a Nikon SMZ1270 stereomicroscope (Nikon) equipped with a Nikon DS-Qi2 monochrome microscope camera and a DS-F 1 × F-mount adapter ×1. Pictures were captured in the bright field channel and with an ED Plan ×1/WF objective at an exposure time of 20 ms, gain 1.2×. NIS-Element AR software (Nikon) was used to generate bright field images. Colony area was measured by binary thresholding (LUTs <15,100).

## Inhibition assays

Antibacterial activities of iturin and linear iturin were tested against *Xanthomonas campestris* and *Agrobacterium tumefaciens*. The activity of iturin and linear iturin was quantified in microtiter plates (96 wells) filled with 250 µL of LB liquid medium, inoculated at OD$_{600\ nm}$ of 0.01 with *X. campestris* or *Agrobacterium tumefaciens* from fresh preculture following the same protocol as for *Bacillus* cultures. The activity of iturin and linear iturin was estimated by measuring the pathogen OD$_{600\ nm}$ every 30 min for 24 h with a Tecan Spark microplate reader, continuously shaken at 150 rpm and at 30°C.

For antifungal activities, we first prepared stock solution of spores. To that end, *Fusarium* and *Botrytis* fungi were grown on potato dextrose agar (PDA) medium (potato extract 4 g/L, dextrose 20 g/L, and agar 15 g/L) plates in the dark for 3 days at room temperature, followed by 1 day at daylight and 3 subsequent days in the dark. Spores were then collected, filtered, and stored at −80°C in peptone water supplemented with glycerol 25% (vol/vol) for *Fusarium* and 50% (vol/vol) for *Botrytis*. Then, antifungal assays were performed following the same protocol as for antibacterial activity. The activities of iturin and linear iturin were quantified in microtiter plates (96 wells) filled with 250 µL

of potato dextrose broth (PDB) liquid medium (PDA without agar), inoculated with $10^6$ spores/mL of *Fusarium* or *Botrytis* from stock spore solutions. The activities of iturin and linear iturin were estimated by measuring the pathogen $OD_{600\,nm}$ every 30 min for 24 h with a Tecan Spark microplate reader, continuously shaken at 150 rpm and at 30°C.

## Proteomic analysis

The comparative proteomic analysis was performed on CFS of *Streptomyces venezuelae* grown on ISP6 (yeast extract 1 g/L, peptone 15 g/L, proteose peptone 5 g/L, ferric ammonium citrate 0.5 g/L, $Na_2S_2O_3$ 0.08 g/L, agar 20 g/L, pH 7.0–7.2), conducive for the degradation of Bv CLPs, versus *Streptomyces venezuelae* grown on ISP5, non-conducive for the degradation of Bv CLPs [L-asparagine 1 g/L, glycerol 10 g/L, $K_2HPO_4$ 1 g/L, 1 mL of trace solution ($MnCl_2 \cdot 4H_2O$ 0.01 g/L, $FeSO4 \cdot 7H_2O$ 0.01 g/L, $ZnSO_4 \cdot 7H_2O$ 0.01 g/L), agar 20 g/L, pH 7.0–7.4]. CFSs were prepared as described in "Generation of cell-free supernatants" section. We did not used the ISP2 medium as conducive medium for comparative proteomics as the CFS generated from ISP2 cultures contained too many residual proteins and peptides from the yeast and malt extracts.

The protein concentration was determined using an RC DC Protein Assay Kit (Bio-Rad, USA). A total of 10 μg of proteins was transferred into Eppendorf tubes then reduced, alkylated, and reduced prior to the purification with the 2D Clean-Up kit (GE Healthcare Life Sciences, USA), following the guidelines provided by the manufacturer. The protein precipitates were dissolved in a solution containing 50-mM bicarbonate ammonium before being digested for 16 h at 37°C using a trypsin solution [trypsin:total protein ratio (wt/wt), of 1:50]. Following this step, a fivefold dilution was performed using acetonitrile (at 80% vol/vol), followed by a 3-h incubation at 37°C with the addition of fresh trypsin [trypsin/total protein ratio (wt/wt) of 1:100]. The digested samples were dried by speed vacuum, and an aliquot corresponding to 3.5 μg of digested proteins was purified using a Zip-Tip C18 (Millipore, Billerica, USA). The generated peptides were evaporated to dryness in a speed vacuum, conditioned at 1.0 μg/9 μL in ammonium formate of 100 mM and spiked with either MassPREP Digestion Standard (MPDS) mix 1 or MPDS mix 2 for data comparison.

The LC-MS/MS and data analysis were performed as follows. The separation of digested peptides was carried out through reversed-phase chromatography utilizing a nanoACQUITY M-Class UPLC system (Waters Corporation) connected to a Q Exactive Plus (Thermo Fisher Scientific, Waltham, MA, USA) equipped with a nanospray source. The trap column was a symmetry C18 5 μm (180 μm × 20 mm), and the analytical column was an HSS T3 C18 1.8 μm (75 μm × 250 mm) (Waters). One microgram of the samples was loaded at 20 μL/min on the trap column in 98% solvent A for 3 min and subsequently separated on the analytical column at a flow rate of 500 nL/min with a linear gradient. Solvent A is 0.1% formic acid in water, and solvent B is 0.1% formic acid in acetonitrile. The total run time is 180 min.

The mass spectrometry techniques employed for all analyses were TopN data-dependent acquisition mode that automatically triggers the MS/MS experiments with a designated N value of 12. In this method, the spectrometer first captures a complete MS spectrum, from which the 12 most intense peaks are selected (excluding those with single charges and unassigned charge precursors).

Regarding the acquisition parameters for MS, they encompass a mass range from 400 to 1,600 *m/z*; a resolution of 70,000; an automatic gain control (AGC) target of $3 \times 10^6$; or a maximum injection time of 200 ms. For precursor ions, the selection windows were 2.0 m/z, the AGC target was set to $1 \times 10^5$ (or 50 ms as a maximum of injection time) and the resolving power of 17,500 at *m/z* of 200. Normalized collision energy was 28. A dynamic exclusion of 10 s was also applied to avoid the redundancy of MS/MS spectra of the same ions.

MaxQuant software (version 1.6.14) was used to analyze raw mass spectrometric files, and identification was ran against the Uniprot *Streptomyces venezualea* ATCC 10712 (downloaded on 25 April 2023). Perseus software (version 1.6.10) was then used to

perform further downstream statistical and bioinformatic analyses of the MaxQuant processing results.

## ACKNOWLEDGMENTS

We warmly thank Guillaume Balleux and Pascale Bonnet for proofreading this article. We also gratefully acknowledge Sébastien Steels and Catherine Helmus for their technical support.

This work was supported by Action de Recherche Concertée, Mind Project, from the University of Liège, by the EU Interreg V France-Wallonie-Vlaanderen portfolio Smart-Biocontrol (Bioscreen and Bioprotect projects, avec le soutien du Fonds Européen de Développement Régional - Met steun van het Europees Fonds voor Regionale Ontwikkeling), the European Union Horizon 2020 research and innovation program under grant agreement no. 731077 and by the EOS project ID 30650620 from the FWO/F.R.S.-FNRS. A.R. and A.A. are recipients of a F.R.I.A. fellowship (F.R.S.-FNRS, National Funds for Scientific Research in Belgium), and M.O. is research director at the F.R.S.-FNRS. S.R. is a Senior Research Associate at the Belgian Fund for Scientific Research (F.R.S.-FNRS, Brussels, Belgium). The Q-Exactive mass spectrometer was funded by ERDF (European Regional Development Fund) and the Walloon Region grants: BIOMED HUB Technology Support (# 2.2.1/996). The authors are grateful for the financial support from the University of Liège by means of the Concerted Action Research MIND (2020–2023).

A.R. did most of the experiments; A.R. and A.A.A did the LC-MS(/MS) analysis and metabolomics; A.A. and A.R. did the stereomicroscopic analysis; P.B. did the proteomic experiments; A.R. and M.O. mainly wrote the article; S.R. and L.Q. substantially revised the article and were involved in the discussion of the work; M.O. supervised the study. All of the authors commented on the article and contributed to the final form.

## AUTHOR AFFILIATIONS

[1]Microbial Processes and Interactions laboratory, TERRA teaching and research centre, Gembloux Agro-Bio Tech,University of Liège, Gembloux, Belgium
[2]Department of Chemistry, University of Liège, Liège, Belgium
[3]InBioS—Centre for Protein Engineering,University of Liège, Liege, Belgium
[4]Hedera-22, Liege, Belgium

## AUTHOR ORCIDs

Augustin Rigolet http://orcid.org/0000-0001-7015-8354
Sébastien Rigali http://orcid.org/0000-0003-4022-7325
Marc Ongena http://orcid.org/0000-0002-7418-4354

## ADDITIONAL FILES

The following material is available online.

### Supplemental Material

**Supplemental material (Spectrum03106-S0001.docx).** Fig. S1 to S13 and Tables S1 to S4.

### Open Peer Review

**PEER REVIEW HISTORY (review-history.pdf).** An accounting of the reviewer comments and feedback.

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
