## [Reviewer comments · Microbiology Spectrum]

Microbiology Spectrum

Lipopeptides as rhizosphere public goods for microbial cooperation

Augustin Rigolet, Anthony Argüelles Arias, Adrien Anckaert, Loïc Quinton, Sébastien Rigali, Deborah Tellatin, Pierre Burguet, and Marc Ongena

Corresponding Author(s): Augustin Rigolet, Universite de Liege Gembloux Agro-Bio Tech

Review Timeline:

Submission Date:	August 16, 2023
Editorial Decision:	September 11, 2023
Revision Received:	September 27, 2023
Editorial Decision:	October 30, 2023
Revision Received:	November 1, 2023
Accepted:	November 6, 2023

Editor: Eva Sonnenschein

Reviewer(s): Disclosure of reviewer identity is with reference to reviewer comments included in decision letter(s). The following individuals involved in review of your submission have agreed to reveal their identity: Ling Ding (Reviewer #1); Thomas Rey (Reviewer #2)

Transaction Report:

DOI: <https://doi.org/10.1128/spectrum.03106-23>

September 11, 2023

Mr. Augustin Rigolet
Universite de Liege Gembloux Agro-Bio Tech
Microbial Processes and Interactions laboratory, TERRA teaching and research centre
Avenue de la faculté d'Agronomie
Gembloux, Namur 5030
Belgium

Re: Spectrum03106-23 (Lipopeptides as rhizosphere public goods for microbial cooperation)

Dear Mr. Augustin Rigolet:

Link Not Available

Sincerely,

Eva Sonnenschein

Journals Department
Reviewer comments:

Reviewer #1 (Comments for the Author):

The authors have investigated the degradation of several *Streptomyces*, especially *S. venezuelae* in hydrolysis and degradation of cyclic peptides by *Bacillus velezensis*.

1. While the EICs using HRMS can give a qualitative aspect for presence/absence of secondary metabolites, one cannot use the peak area to reflect the quantity. Moreover, the biomass is different from the confrontation and the Bv side. MALDI imaging is better suited for visualization of distribution of different CLPs and their degradation products.
2. The rhizosphere model is rather superficial. On one hand, most *Streptomyces* including *S. venezuelae* are not reported to colonize roots. They exist mostly in soil. On the other hand, the authors have added rather nutritious media components into the

root system. One must be careful with the conclusion that "lipopeptides as rhizosphere public goods for microbial cooperation". Please make a real rhizosphere experiment using pure root extrudate. Or please use a soil system to give a conclusion.

Reviewer #2 (Comments for the Author):

This research article from Rigolet et al. work on an emerging research field in plant-microbes interaction which is the intermicrobial communication within the rhizosphere. Microbial specialised metabolites have been investigated from a medical and biotechnological perspectives for more than 70 years, however their ecological relevance in terms of fitness for the strains that produce them has been largely neglected. Here the authors combined outstanding microscopic, microbial genetic and cutting-edge metabolomic approaches to establish how cyclic lipopeptides from bacillus can be degraded by a yet to discover *Streptomyces* peptidase. The authors showed that CLPs degradation into linear form and subsequent degradation product lead to reduced motility of bacillus and reduced antifungal activity. Importantly, using polyketides biosynthesis deficient strain of *Bacillus*, they could show that the degraded forms of CLP is likely to be taken up an N source by the *Streptomyces*. Whether this behavior can happen in natural ecosystem with wild type *Bacillus* strain remain to be addressed.

The standards of the figures and methods used in this research are excellent. The manuscript is extremely clear and well written. The conclusions drawn from the experiments are completely appropriate and constitute a crucial milestone in our understanding of intermicrobial communication mediated by specialised metabolites.

Here a list of points of improvement for the manuscript :

MAJOR POINTS :

-Figure 1 and Figure 2 are uploaded in the wrong order in the MS

MINOR POINTS :

-The supplementary material used different polices and sizes, please uniformise

-I have the feeling that methods regarding extraction and sample preparation for CLPs analysis should be regrouped in a dedicated paragraph instead of being split between microbiological and in planta experiments

-Line 92 and afterward : It can be noticed from figure 1D that the *Streptomyces* pellet surrounding the root turn green in presence of *Bacillus*. It is worth mentioning explicitly that this is because it is colonised by GFP-tagged *Bacillus*.

- I have the feeling that the article title could be more explicit and state clearly that degraded CLP can be used by *Streptomyces* as N source

Staff Comments:

Preparing Revision Guidelines

Please return the manuscript within 60 days; if you cannot complete the modification within this time period, please contact me. If you do not wish to modify the manuscript and prefer to submit it to another journal, please notify me of your decision immediately so that the manuscript may be formally withdrawn from consideration by Microbiology Spectrum.

Reviewer #1 (Comments for the Author):

R1 (1). The authors have investigated the degradation of several *Streptomyces*, especially *S. venezuelae* in hydrolysis and degradation of cyclic peptides by *Bacillus velezensis*.

1. While the EICs using HRMS can give a qualitative aspect for presence/absence of secondary metabolites, one cannot use the peak area to reflect the quantity. Moreover, the biomass is different from the confrontation and the Bv side. MALDI imaging is better suited for visualization of distribution of different CLPs and their degradation products.

Our response: We understand the comment. Of course peak areas obtained in any LC-MS analysis of biological extracts do not give access to absolute quantification of compounds unless standards of the highest purity are available in sufficient amounts for calibration curves. This is not the case for lipopeptide degradation products, but our goal in this study was not to precisely quantify these products or their linear/cyclic precursors but provide an idea of the relative accumulation of all these products. What we wanted to decode is the mechanism of degradation, which can be deduced from the sequential appearance/disappearance of the intermediates resulting from incubation of the CLPs in presence of *Streptomyces* enzymes. In that respect, even if their precise amounts may be unknown, what can be deduced from the kinetic of accumulation of degradation products is still valid according to relative quantification based on peak area. Moreover, prior to any experiment, we validated the extraction and relative quantification method. We mixed different dilutions of *B. velezensis* cell free extract with agar (14 g/l), re-extracted the metabolites as described in the Material and Methods and analyzed the resulting extracts with our in house-optimized LC-MS-MS method. You will find in the figure here below the calibration curves that confirm the linearity between the peak area and the relative concentration of each CLP as added in the agar. It indicates that this method is suitable for relative quantification. Moreover, we designed our plate experiments to overcome possible limitations and bias due to inconsistent sampling. We standardized the agar plugs and the distance of these plugs from the *Bacillus* colonies considering a homogenous diffusion of the compounds. We modified the material and methods to better clarify the extraction procedure.

Regarding the difference of biomass, most of the difference as seen on the picture (flat white powdery veil) is due to sliding of the colony. According to recent studies on biofilm and labor division, BSMs producers form distinct subpopulation with motile cell and matrix producers. Thereby, the biomass of sliding colonies mostly gather motile cells that do not participate much in the production of the CLPs (DOI: 10.1128/microbiolspec.mb-0002-2014 and DOI: 10.1002/pmic.200701025).

R1 is right, MALDI imaging would indeed have offered a higher spatial resolution of the different

metabolites. In previous studies, we successfully exploited this technology for spatio-temporal monitoring of bacterial metabolites from *Bacillus* in interaction with plant or other microbes (DOI: 10.1021/ac500290s, DOI: 10.1111/1751-7915.12238, DOI : 10.1111/1758-2229.12286, DOI: 10.1128/spectrum.02038-21). We also attempted to use it here to map degradation products, but unfortunately and as illustrated in the following figure for iturin, we could only hardly detect the linearized form of the lipopeptide and one degradation product in the interaction zone. In a way, it confirms our results from the UPLC MS-MS analyses of solid medium extracts, but it shows limitations of the IMS in this particular case where the other and shorter degradation products could not be detected despite using various settings to improve extraction of these smaller ions .

R1 (2). The rhizosphere model is rather superficial. On one hand, most *Streptomyces* including *S. venezuelae* are not reported to colonize roots. They exist mostly in soil.

Our response: We strongly disagree with this comment. The idea here is to study interaction between *B. velezensis* as plant-associated rhizobacterium and *Streptomyces* species which are described to dwell not only in soil but have been isolated in many instances from the rhizosphere and other ecological niches. It is well established that *Streptomyces* do develop intricate interactions with plants (notably by sensing host signals to activate the production of bioactive molecules) and, such as bacilli, are common members of the plant-associated microbiome (DOI: 10.1146/annurev-arplant-050312-120106 , DOI: 10.1038/ismej.2008.80, DOI: 10.1007/s10482-018-1014-z, DOI: 10.1038/nature11237). It is thus clear that some *Streptomyces* species can dwell in a rhizosphere-adapted life style and display efficient root colonization potential. More specifically, a recent paper showed that the strain *S. venezuelae* ATCC10712 used in this study, should not be considered as a simple soil bacteria but also as a PGPR since it promotes plant health, alleviate salt stress and has been found associated with plant roots (<http://dx.doi.org/10.1038/s41579-020-0379-y>). The fact that a given bacterial species can be found both in the bulk soil and in the rhizosphere compartment is far from being aberrant since those two niches overlap and form a continuum (DOI: 10.1073/pnas.1414592112). By the way, it is exactly what our plate experiments shows since Sv readily grows on root exudates and our *in planta* experiments further supports that Sv can sustain some growth in the root vicinity.

R1 (3) On the other hand, the authors have added rather nutritious media components into the root system. One must be careful with the conclusion that "lipopeptides as rhizosphere public

goods for microbial cooperation". Please make a real rhizosphere experiment using pure root exudate. Or please use a soil system to give a conclusion.

Our response: As the reviewer may know, the study of metabolite-driven interspecies interactions in real conditions is extremely challenging due to the difficulty of reliable extraction of these small-size chemicals from the complex soil matrix. This is well known and particularly true for charged amphiphilic compounds such as cyclic lipopeptides, which tightly adsorb to soil particles. Attempts to profile these secondary metabolites as produced in the "real" rhizosphere soil usually fail and it is widely recognized that insights can only be gained by using *in vitro* "simplified" systems. That said, most studies on microbial interactions use rich culture media that do not reflect the nutritional context of the natural habitat, i.e. the rhizosphere. Here we deliberately wanted to use culture media which composition somehow reflects the natural oligotrophic rhizosphere environment thereby both ensuring conditions in which the two bacteria have successfully evolved and the presence of elements for which their genetic material has been shaped for utilization and therefore also for subsequent metabolic response. We performed most experiments using a root exudate-mimicking medium especially to address this concern. It is a synthetic medium that mimics the composition in main chemicals released by roots of Solanaceae plants including tomato and that has been already used (and thus "validated") in previous works (DOI: 10.1038/s41396-022-01337-1 , DOI: 10.1128/spectrum.02038-21). Using pure naturally produced root exudates wouldn't have been more representative since their chemical nature and relative abundance widely vary according to many factors such as the manipulation of the plants, the plant cultivar/species, the use of hydroponics to recover the exudates, the growth in sterile conditions etc...

Still, we are aware about the limitation of this *in vitro* set up and it's why we performed the *in planta* experiments. In these assays, we used the standard minimal Hoagland medium as substrate to grow plants, which contains limited amounts of nitrogen (15 μ M) but no carbon source (<https://doi.org/10.1016/j.molp.2023.06.001>, <https://doi.org/10.1111/tpj.12995>). As preliminary test, we verified that neither bacteria could grow on this medium alone and cannot therefore use agar as carbon source. So in such setting, bacterial development and root colonization can only be supported by the consumption of exudates naturally secreted by tomato plants. Maybe the way we describe the methodology was not clear but the root system we used is thus completely free of any "additional nutritious components".

Regarding the title "lipopeptides as rhizosphere public goods for microbial cooperation", see response to reviewer 2

Reviewer #2 (Comments for the Author):

This research article from Rigolet et al. work on an emerging research field in plant-microbes interaction which is the intermicrobial communication within the rhizosphere. Microbial specialised metabolites have been investigated from a medical and biotechnological perspectives for more than 70 years, however their ecological relevance in terms of fitness for the strains that produce them has been largely neglected. Here the authors combined outstanding microscopic, microbial genetic and cutting-edge metabolomic approaches to establish how cyclic lipopeptides from bacillus can be degraded by a yet to discover Streptomyces peptidase. The authors showed

that CLPs degradation into linear form and subsequent degradation product lead to reduced motility of bacillus and reduced antifungal activity. Importantly, using polyketides biosynthesis deficient strain of Bacillus, they could show that the degraded forms of CLP is likely to be taken up an N source by the Streptomyces. Whether this behavior can happen in natural ecosystem with wild type Bacillus strain remain to be addressed.

The standards of the figures and methods used in this research are excellent. The manuscript is extremely clear and well written. The conclusions drawn from the experiments are completely appropriate and constitute a crucial milestone in our understanding of intermicrobial communication mediated by specialised metabolites.

Our response: We warmly thank the reviewer for the positive feedback and interest in our work!

Here a list of points of improvement for the manuscript :

MAJOR POINTS :

-Figure 1 and Figure 2 are uploaded in the wrong order in the MS

Our response: We apology for this mistake and corrected this in the new version.

MINOR POINTS :

-The supplementary material used different polices and sizes, please uniformise

Our response: Thanks, sorry for that, it is consistent now in the revised version.

-I have the feeling that methods regarding extraction and sample preparation for CLPs analysis should be regrouped in a dedicated paragraph instead of being split between microbiological and in planta experiments

Our response: We agree and re-worked the M&M accordingly.

-Line 92 and afterward : It can be noticed from figure 1D that the Streptomyces pellet surrounding the root turn green in presence of Bacillus. It is worth mentioning explicitly that this is because it is colonised by GFP-tagged Bacillus.

Our response: Nothing indicates that *Streptomyces* is colonized by *Bacillus*. It was not thoroughly investigated but clearly, *Streptomyces* colonies are auto-fluorescent as also observed when inoculated alone on tomato roots (doi: 10.3791/53863) This is explicitly mentioned in the legend of the figure in the revised version. Therefore, based on the images, we assume that the two bacteria segregate on the root system. Still the two species can develop close to each other and thus in a way that diffusing CLPs may be degraded by Sv secreted enzymes in the surrounding medium.

- I have the feeling that the article title could be more explicit and state clearly that degraded CLP can be used by Streptomyces as N source

Our response: In this article, we provide evidence that *Streptomyces* catabolizes the CLPs of *Bacillus* and use it to sustain its growth. Current knowledge of the nutritional context of the rhizosphere indicates that nutrients are scarce in this environment and that the main source of nutrient is the root exudates. The composition of those root exudates can vary a lot according to the plant genotype/species, the plant health status, the soil composition, the microbiome, etc. Nevertheless, its main constituents are sugars and organic acids and Nitrogen containing compounds are fairly rare in comparison. Oppositely, CLPs are nitrogen rich compounds (thanks to the peptide ring) that are readily produced by many root-associated bacteria such as *Bacillus*. We also show that their degradation generates free amino acids. Hence, we proposed the nutrition we observed in our experiments is due to the fueling of the nitrogen metabolism of *Streptomyces*. However, we do not provide direct evidence for this specific aspect of N nutrition of *Streptomyces*. Consequently, we believe that, although it is a clear and interesting direct perspective of this work, it is still too speculative and we didn't want to put emphasis on it in the title and prefer avoiding a statement like the one proposed "degraded CLP can be used by *Streptomyces* as N source". We would prefer to keep with a more "global" reference to nutrition strictly relying on our data but, of course, we are ready to adapt according to further discussion with reviewers and the editor if they judge any other title more appropriate.

Re: Spectrum03106-23R1 (Lipopeptides as rhizosphere public goods for microbial cooperation)

Dear Mr. Augustin Rigolet:

Thank you for the privilege of reviewing your work. Your article is very close to acceptance. As a final modification, could I please ask you to move the method section from the supplementary material into the main manuscript?

Revision Guidelines

Sincerely,
Eva Sonnenschein
Editor
Microbiology Spectrum

Reviewer #1 (Comments for the Author):

R1 (1). The authors have investigated the degradation of several *Streptomyces*, especially *S. venezuelae* in hydrolysis and degradation of cyclic peptides by *Bacillus velezensis*.

1. While the EICs using HRMS can give a qualitative aspect for presence/absence of secondary metabolites, one cannot use the peak area to reflect the quantity. Moreover, the biomass is different from the confrontation and the Bv side. MALDI imaging is better suited for visualization of distribution of different CLPs and their degradation products.

Our response: We understand the comment. Of course peak areas obtained in any LC-MS analysis of biological extracts do not give access to absolute quantification of compounds unless standards of the highest purity are available in sufficient amounts for calibration curves. This is not the case for lipopeptide degradation products, but our goal in this study was not to precisely quantify these products or their linear/cyclic precursors but provide an idea of the relative accumulation of all these products. What we wanted to decode is the mechanism of degradation, which can be deduced from the sequential appearance/disappearance of the intermediates resulting from incubation of the CLPs in presence of *Streptomyces* enzymes. In that respect, even if their precise amounts may be unknown, what can be deduced from the kinetic of accumulation of degradation products is still valid according to relative quantification based on peak area. Moreover, prior to any experiment, we validated the extraction and relative quantification method. We mixed different dilutions of *B. velezensis* cell free extract with agar (14 g/l), re-extracted the metabolites as described in the Material and Methods and analyzed the resulting extracts with our in house-optimized LC-MS-MS method. You will find in the figure here below the calibration curves that confirm the linearity between the peak area and the relative concentration of each CLP as added in the agar. It indicates that this method is suitable for relative quantification. Moreover, we designed our plate experiments to overcome possible limitations and bias due to inconsistent sampling. We standardized the agar plugs and the distance of these plugs from the *Bacillus* colonies considering a homogenous diffusion of the compounds. We modified the material and methods to better clarify the extraction procedure.

Regarding the difference of biomass, most of the difference as seen on the picture (flat white powdery veil) is due to sliding of the colony. According to recent studies on biofilm and labor division, BSMs producers form distinct subpopulation with motile cell and matrix producers. Thereby, the biomass of sliding colonies mostly gather motile cells that do not participate much in the production of the CLPs (DOI: 10.1128/microbiolspec.mb-0002-2014 and DOI: 10.1002/pmic.200701025).

R1 is right, MALDI imaging would indeed have offered a higher spatial resolution of the different

metabolites. In previous studies, we successfully exploited this technology for spatio-temporal monitoring of bacterial metabolites from *Bacillus* in interaction with plant or other microbes (DOI: 10.1021/ac500290s, DOI: 10.1111/1751-7915.12238, DOI : 10.1111/1758-2229.12286, DOI: 10.1128/spectrum.02038-21). We also attempted to use it here to map degradation products, but unfortunately and as illustrated in the following figure for iturin, we could only hardly detect the linearized form of the lipopeptide and one degradation product in the interaction zone. In a way, it confirms our results from the UPLC MS-MS analyses of solid medium extracts, but it shows limitations of the IMS in this particular case where the other and shorter degradation products could not be detected despite using various settings to improve extraction of these smaller ions .

R1 (2). The rhizosphere model is rather superficial. On one hand, most *Streptomyces* including *S. venezuelae* are not reported to colonize roots. They exist mostly in soil.

Our response: We strongly disagree with this comment. The idea here is to study interaction between *B. velezensis* as plant-associated rhizobacterium and *Streptomyces* species which are described to dwell not only in soil but have been isolated in many instances from the rhizosphere and other ecological niches. It is well established that *Streptomyces* do develop intricate interactions with plants (notably by sensing host signals to activate the production of bioactive molecules) and, such as bacilli, are common members of the plant-associated microbiome (DOI: 10.1146/annurev-arplant-050312-120106 , DOI: 10.1038/ismej.2008.80, DOI: 10.1007/s10482-018-1014-z, DOI: 10.1038/nature11237). It is thus clear that some *Streptomyces* species can dwell in a rhizosphere-adapted life style and display efficient root colonization potential. More specifically, a recent paper showed that the strain *S. venezuelae* ATCC10712 used in this study, should not be considered as a simple soil bacteria but also as a PGPR since it promotes plant health, alleviate salt stress and has been found associated with plant roots (<http://dx.doi.org/10.1038/s41579-020-0379-y>). The fact that a given bacterial species can be found both in the bulk soil and in the rhizosphere compartment is far from being aberrant since those two niches overlap and form a continuum (DOI: 10.1073/pnas.1414592112). By the way, it is exactly what our plate experiments shows since *Sv* readily grows on root exudates and our *in planta* experiments further supports that *Sv* can sustain some growth in the root vicinity.

R1 (3) On the other hand, the authors have added rather nutritious media components into the root system. One must be careful with the conclusion that "lipopeptides as rhizosphere public

goods for microbial cooperation". Please make a real rhizosphere experiment using pure root exudate. Or please use a soil system to give a conclusion.

Our response: As the reviewer may know, the study of metabolite-driven interspecies interactions in real conditions is extremely challenging due to the difficulty of reliable extraction of these small-size chemicals from the complex soil matrix. This is well known and particularly true for charged amphiphilic compounds such as cyclic lipopeptides, which tightly adsorb to soil particles. Attempts to profile these secondary metabolites as produced in the "real" rhizosphere soil usually fail and it is widely recognized that insights can only be gained by using *in vitro* "simplified" systems. That said, most studies on microbial interactions use rich culture media that do not reflect the nutritional context of the natural habitat, i.e. the rhizosphere. Here we deliberately wanted to use culture media which composition somehow reflects the natural oligotrophic rhizosphere environment thereby both ensuring conditions in which the two bacteria have successfully evolved and the presence of elements for which their genetic material has been shaped for utilization and therefore also for subsequent metabolic response. We performed most experiments using a root exudate-mimicking medium especially to address this concern. It is a synthetic medium that mimics the composition in main chemicals released by roots of Solanaceae plants including tomato and that has been already used (and thus "validated") in previous works (DOI: 10.1038/s41396-022-01337-1 , DOI: 10.1128/spectrum.02038-21). Using pure naturally produced root exudates wouldn't have been more representative since their chemical nature and relative abundance widely vary according to many factors such as the manipulation of the plants, the plant cultivar/species, the use of hydroponics to recover the exudates, the growth in sterile conditions etc...

Still, we are aware about the limitation of this *in vitro* set up and it's why we performed the *in planta* experiments. In these assays, we used the standard minimal Hoagland medium as substrate to grow plants, which contains limited amounts of nitrogen (15 μ M) but no carbon source (<https://doi.org/10.1016/j.molp.2023.06.001>, <https://doi.org/10.1111/tpj.12995>). As preliminary test, we verified that neither bacteria could grow on this medium alone and cannot therefore use agar as carbon source. So in such setting, bacterial development and root colonization can only be supported by the consumption of exudates naturally secreted by tomato plants. Maybe the way we describe the methodology was not clear but the root system we used is thus completely free of any "additional nutritious components".

Regarding the title "lipopeptides as rhizosphere public goods for microbial cooperation", see response to reviewer 2

Reviewer #2 (Comments for the Author):

This research article from Rigolet et al. work on an emerging research field in plant-microbes interaction which is the intermicrobial communication within the rhizosphere. Microbial specialised metabolites have been investigated from a medical and biotechnological perspectives for more than 70 years, however their ecological relevance in terms of fitness for the strains that produce them has been largely neglected. Here the authors combined outstanding microscopic, microbial genetic and cutting-edge metabolomic approaches to establish how cyclic lipopeptides from bacillus can be degraded by a yet to discover Streptomyces peptidase. The authors showed

that CLPs degradation into linear form and subsequent degradation product lead to reduced motility of bacillus and reduced antifungal activity. Importantly, using polyketides biosynthesis deficient strain of Bacillus, they could show that the degraded forms of CLP is likely to be taken up an N source by the Streptomyces. Whether this behavior can happen in natural ecosystem with wild type Bacillus strain remain to be addressed.

The standards of the figures and methods used in this research are excellent. The manuscript is extremely clear and well written. The conclusions drawn from the experiments are completely appropriate and constitute a crucial milestone in our understanding of intermicrobial communication mediated by specialised metabolites.

Our response: We warmly thank the reviewer for the positive feedback and interest in our work!

Here a list of points of improvement for the manuscript :

MAJOR POINTS :

-Figure 1 and Figure 2 are uploaded in the wrong order in the MS

Our response: We apology for this mistake and corrected this in the new version.

MINOR POINTS :

-The supplementary material used different polices and sizes, please uniformise

Our response: Thanks, sorry for that, it is consistent now in the revised version.

-I have the feeling that methods regarding extraction and sample preparation for CLPs analysis should be regrouped in a dedicated paragraph instead of being split between microbiological and in planta experiments

Our response: We agree and re-worked the M&M accordingly.

-Line 92 and afterward : It can be noticed from figure 1D that the Streptomyces pellet surrounding the root turn green in presence of Bacillus. It is worth mentioning explicitly that this is because it is colonised by GFP-tagged Bacillus.

Our response: Nothing indicates that *Streptomyces* is colonized by *Bacillus*. It was not thoroughly investigated but clearly, *Streptomyces* colonies are auto-fluorescent as also observed when inoculated alone on tomato roots (doi: 10.3791/53863) This is explicitly mentioned in the legend of the figure in the revised version. Therefore, based on the images, we assume that the two bacteria segregate on the root system. Still the two species can develop close to each other and thus in a way that diffusing CLPs may be degraded by Sv secreted enzymes in the surrounding medium.

- I have the feeling that the article title could be more explicit and state clearly that degraded CLP can be used by Streptomyces as N source

Our response: In this article, we provide evidence that *Streptomyces* catabolizes the CLPs of *Bacillus* and use it to sustain its growth. Current knowledge of the nutritional context of the rhizosphere indicates that nutrients are scarce in this environment and that the main source of nutrient is the root exudates. The composition of those root exudates can vary a lot according to the plant genotype/species, the plant health status, the soil composition, the microbiome, etc. Nevertheless, its main constituents are sugars and organic acids and Nitrogen containing compounds are fairly rare in comparison. Oppositely, CLPs are nitrogen rich compounds (thanks to the peptide ring) that are readily produced by many root-associated bacteria such as *Bacillus*. We also show that their degradation generates free amino acids. Hence, we proposed the nutrition we observed in our experiments is due to the fueling of the nitrogen metabolism of *Streptomyces*. However, we do not provide direct evidence for this specific aspect of N nutrition of *Streptomyces*. Consequently, we believe that, although it is a clear and interesting direct perspective of this work, it is still too speculative and we didn't want to put emphasis on it in the title and prefer avoiding a statement like the one proposed "degraded CLP can be used by *Streptomyces* as N source". We would prefer to keep with a more "global" reference to nutrition strictly relying on our data but, of course, we are ready to adapt according to further discussion with reviewers and the editor if they judge any other title more appropriate.

Re: Spectrum03106-23R2 (Lipopeptides as rhizosphere public goods for microbial cooperation)

Dear Mr. Augustin Rigolet:

Your manuscript has been accepted, and I am forwarding it to the ASM production staff for publication. Your paper will first be checked to make sure all elements meet the technical requirements. ASM staff will contact you if anything needs to be revised before copyediting and production can begin. Otherwise, you will be notified when your proofs are ready to be viewed.

Sincerely,
Eva Sonnenschein
Editor
Microbiology Spectrum